# Brief Communication: Conventional assumptions involving the speed of radar waves in snow introduce systematic underestimates to sea ice thickness and seasonal growth rate estimates

Robbie D.C. Mallett [1], Isobel R. Lawrence [1], Julienne C. Stroeve [1,2,3], Jack C. Landy [4], and Michel Tsamados [1]

[1]Centre for Polar Observation and Modelling, Earth Sciences, University College London, London, UK
[2]National Snow and Ice Data Center, University of Colorado, Boulder, CO, USA
[3]Centre for Earth Observation Science, University of Manitoba, Winnipeg, Canada
[4]School of Geographical Sciences, University of Bristol, Bristol, UK

**Correspondence:** Robbie Mallett (robbie.mallett.17@ucl.ac.uk)

**Abstract.**

Pan-Arctic sea ice thickness has been monitored over recent decades by satellite radar altimeters such as CryoSat-2, which emits Ku-band radar waves that are assumed in publicly available sea ice thickness products to penetrate overlying snow and scatter from the ice-snow interface. Here we examine two expressions for the time delay caused by slower radar wave propagation through the snow layer and related assumptions concerning the time-evolution of overlying snow density. Two conventional treatments introduce systematic underestimates into ice thickness estimates of up to 15 cm and into thermodynamic growth rate estimates of up to 10 cm over multiyear ice in winter. Correcting these biases would impact a wide variety of model projections, calibrations, validations and reanalyses.

## 1 Introduction

Sea ice is a key moderator of the global climate system, limiting the exchange of heat, moisture and momentum between the ocean and the atmosphere. It also plays a crucial role in ocean circulation and Arctic Ocean primary productivity (e.g. Sévellec et al., 2017; Chan et al., 2017). During autumn, open water areas form new ice that can grow thermodynamically by 1.5 to 2.5 m over a winter season. Further deformation and thermodynamic ice growth can lead to thicknesses in excess of 5 m. Today the Arctic is undergoing a period of profound transformation, with the area and thickness of the floating sea ice cover in rapid decline (e.g. Stroeve and Notz, 2018; Kwok, 2018). These changes are driven by a variety of factors including later freeze-ups, earlier melt onsets and increased winter air temperatures (Graham et al., 2017; Stroeve et al., 2018).

As well as being a sensitive indicator of climate change, winter sea ice thickness also functions as a prognostic variable in the polar climate system, affecting the amount and distribution of sea ice that will survive the summer melt season. Accurate knowledge of sea ice thickness is particularly important where data are assimilated into forecasting systems and other complex models which often exhibit sensitive dependence on initial conditions (Day et al., 2014).

Sea ice thickness has been observed through various methods including submarines, ice mass-balance buoys, electromagnetic induction sounding and satellite laser and radar altimetry (e.g. Schweiger, 2017; Kwok, 2018). The CryoSat-2 mission has played a leading role over the last decade, providing radar ranging observations from which the sea ice thickness may be derived (Wingham et al., 2006; Laxon et al., 2013; Tilling et al., 2018). Ku-band radar altimeters such as CryoSat-2 and Sentinel-3 do not directly measure sea ice freeboard, but instead measure 'radar freeboard' through a time-of-flight calculation. The radar freeboard is the difference in radar ranging between the snow-ice interface and the local, instantaneous sea level (assuming perfect radar wave penetration through the snowpack). Since the radar wave speed is reduced in snow, a priori knowledge of the snow depth and density is required to convert the radar freeboard to the true ice freeboard. Following the freeboard calculation, sea ice thickness can then be estimated through the assumption of hydrostatic equilibrium (e.g. Laxon et al., 2003). This again requires a priori knowledge of snow depth and density to account for freeboard reduction due to the weight of overlying snow. The impact of correcting for the weight of overlying snow on sea ice thickness is of comparable magnitude to the correction for slower wave propagation in snow (Supplementary Material Sect. 1).

An important consideration in the conversion of radar freeboard ($F_r$) to ice freeboard ($F_i$) and in turn ice thickness is therefore the time delay due to slower radar pulse propagation in snow (Kwok, 2014). In this study we highlight two different approaches to the calculation of this time delay used in published literature. Correct handling of this time delay has a significant impact on the retrieval of sea ice thickness and volume from radar altimetry, as we show here. This is particularly the case as snow settles and densifies over the winter season.

We further investigate the impact of assuming a fixed snow density throughout winter when calculating this time delay. At present no groups producing publicly available sea ice thickness products from CryoSat-2 factor monthly evolution of snow density into their correction for slower radar wave propagation in snow, despite often including an evolving density in their calculation of the floe's hydrostatic equilibrium. The impact of this assumption is assessed and found to introduce underestimates of the rate of winter thermodynamic sea ice growth, with October-April growth currently being underestimated by over 10 cm over multiyear ice.

## 2   Different Treatments of the Radar Propagation Correction

The correction to the radar range to account for slower radar wave propagation in snow, $\delta h = F_i - F_r$, is often expressed as the product of snow depth, $Z$, and some function of wave velocity in snow, $f(c_s)$ (e.g. Tilling et al., 2018; Kwok, 2014) such that:

$$\delta h = Z \times f(c_s) \tag{1}$$

We now present a short derivation of $f(c_s)$ and thus $\delta h$ through consideration of the extra time taken, $\delta t$, for a radar wave to travel a distance $Z$ through a specified snow depth rather than through free space. The time delay induced by the snow layer is expressed:

$$\delta t = t_{snow} - t_{vacuum} \tag{2}$$

$$\delta t = Z/c_s - Z/c \tag{3}$$

$$\delta t = Z(1/c_s - 1/c) \tag{4}$$

Where $c_s$ the wave speed in snow, and $c$ is the radar wave speed in free space ($3 \times 10^8$ ms$^{-1}$). To convert this time delay ($\delta t$) into a path difference ($\delta h$), one multiplies by the speed of the wave in free space:

$$\delta h = \delta t \times c = Z(c/c_s - 1) \tag{5}$$

Some works (Tilling et al., 2018; Kwok and Markus, 2018; Kwok and Kacimi, 2018) have used this formulation to correct the radar range for the slower wave propagation speed through snow. Other works have used an alternative form of Eq. (5), generated by multiplying $\delta t$ in Eq. (4) by the wave speed in snow (Kwok, 2014; Kurtz et al., 2014; Kwok and Cunningham, 2015; Ricker et al., 2015; Armitage and Ridout, 2015; Hendricks et al., 2016; Landy et al., 2017; Xia and Xie, 2018):

$$\delta h = Z_r(1 - c_s/c) \tag{6}$$

For Eq. (6) to be true, $Z_r$ must be regarded as:

$$Z_r = Z(c/c_s) \tag{7}$$

However, $Z_r$ is conventionally interpreted as the real snow depth ($Z$) and $\delta h$ is therefore erroneously reduced by a factor of $c_s/c$. When Eq. (7) is incorporated into Eq. (6), $\delta h$ is redefined in terms of Z and becomes Eq. (5).

Conventional interpretation of $Z_r$ as the real snow depth therefore leads to a bias in the freeboard ($B_f$) where:

$$B_f = Z \times \frac{(c-c_s)^2}{c \times c_s} \tag{8}$$

Bias in the freeboard then propagates into estimates of sea ice thickness by a multiplicative factor of $\rho_w/(\rho_w - \rho_i)$, where $\rho_w$ represents the density of seawater and $\rho_i$ represents the density of sea ice. Because first year ice (FYI) is generally denser than multiyear ice (MYI), a fixed snow thickness will introduce a greater bias on the thickness of first year ice. However, typical biases introduced by this treatment over FYI are generally expected to be lower due to reduced snow accumulation. The bias introduced to sea ice thickness retrievals ($B_{SIT}$) due to conventional, erroneous use of Eq. (6) is therefore:

$$B_{SIT} = Z \times \frac{(c-c_s)^2}{c \times c_s} \times \frac{\rho_w}{\rho_w - \rho_i} \tag{9}$$

Equation (9) illustrates that the bias grows linearly with snow depth. In addition to this, $B_{SIT}$ is also dependent on the speed of the radar wave in snow, which is itself a function of snow density. Several empirical relationships have been proposed for the relationship between snow density and radar wave speed, however the most commonly used three (Hallikainen et al., 1982; Tiuri et al., 1984; Ulaby et al., 1986) deviate negligibly from each other in the typical density range for snow observed on Arctic sea ice (Fig. S1). In this investigation, we use the relationship from Ulaby et al. (1986):

$$c_s = c(1 + 0.51\rho_s)^{-1.5} \tag{10}$$

As snow density increases, $c_s$ decreases and $B_{SIT}$ increases. This positive relationship between $f(c_s)$ and snow density is shown in Fig. (1a). Because both snow depth and snow density generally increase throughout the season as snow accumulates, compacts and settles, any $\delta h$ generated through incorrect expression of $f(c_s)$ becomes increasingly underestimated.

Furthermore, $B_{SIT}$ increases even as a fixed snow water equivalent densifies and shrinks in volume. This is because $B_{SIT}$ scales more rapidly with increasing snow density than it reduces with decreasing snow depth. The increase in bias with snow density for constant SWE is illustrated in Fig. (1b).

Since $B_{SIT}$ is explicitly a function of snow depth and implicitly a function of snow density via Eq. (10), its spatial mapping requires the use of an Arctic snow distribution. Here we use snow depths and densities from Warren et al. (1999) (henceforth 'W99') to illustrate these underestimates. To be consistent with current data products that rely on W99 for their snow depth distribution, we halve snow depths over first-year ice as per Laxon et al. (2013) and only consider the Central Arctic basin (see Fig. S2) where W99 is considered most reliable (Kwok and Cunningham, 2015). Data on sea ice type and extent were taken from the sea ice type product of the EUMETSAT Ocean and Sea Ice Satellite Application Facility (OSISAF; Aaboe et al., 2016).

We find that where sea ice thicknesses are calculated using W99 snow depths and densities in the Central Arctic, thickness underestimates introduced by erroneous interpretation of Eq. (6) increase throughout the winter to values exceeding 15 cm in April over multi-year ice (Fig. 1c). Over FYI the mean bias increases from 4.2 cm in October to 9.8 cm in April (compared to 6.4 cm and 13.6 cm for MYI). In April, 28% of MYI has a bias exceeding 15 cm, and 7% exceeds 16 cm.

How does this bias impact sea ice thickness products currently available to the science community? Most commonly-used products do not correct for slower wave speed using the W99 density distributions in time or space, but instead use a reference density to calculate a fixed value for $f(c_s)$ in Eq. (1). This value is fixed not only across the Arctic basin, but throughout the winter. In the CryoSat-2 sea ice thickness product from the Alfred Wegener Institute (AWI; Hendricks et al., 2016), $f(c_s)$ is taken as $(1 - c_s/c)$ as in Eq. (6). Citing the reference spring snow density given by Kwok (2014) of 350 kg m$^{-3}$, they generate a fixed $\delta h$ of $0.22Z$.

On the other hand, the Centre for Polar Observation and Modelling (CPOM) takes $f(c_s)$ to be $(c/c_s - 1)$ (Eq. (5); Tilling et al., 2018). However, the CPOM product uses a lower reference density of 300 kg m$^{-3}$ (taken from Kwok et al. (2011)), generating a reference $\delta h$ of $0.25Z$. AWI's use of a higher reference density mitigates the difference introduced by their

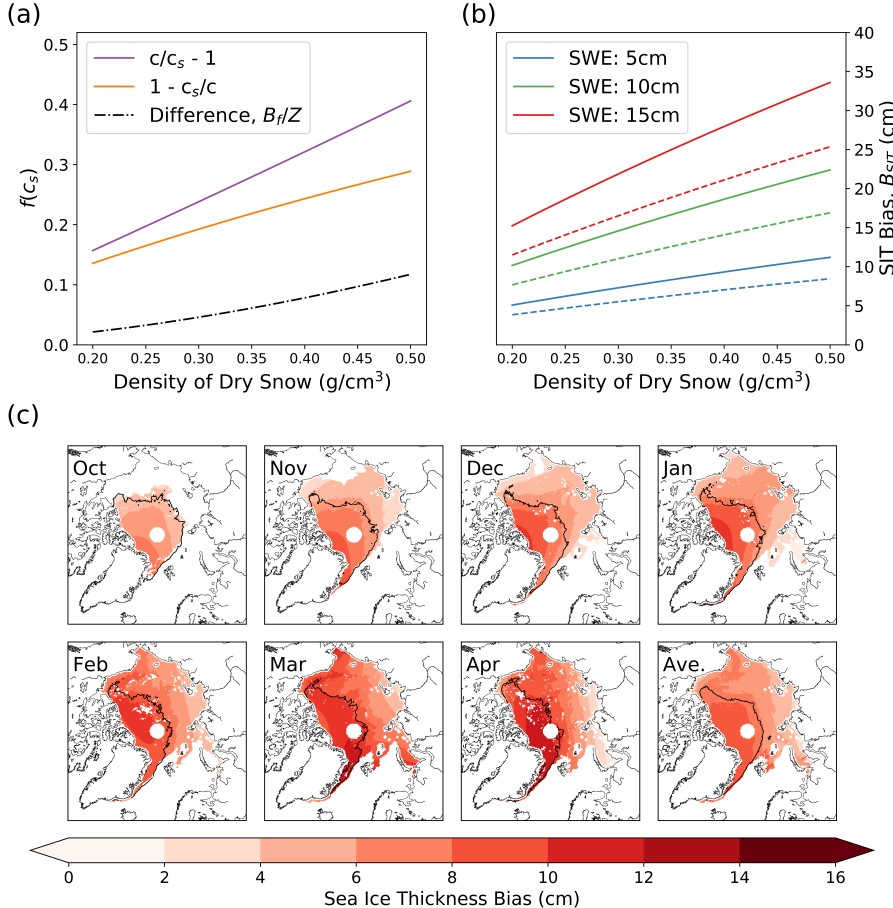

**Figure 1.** (a) Difference between conventional use of Eq. (6) and Eq. (5) as a function of snow density. This bias increases with snow density, ultimately exceeding a factor of 0.1 of the snow depth for dense snow. (b) Sea ice thickness bias for a fixed mass of snow increases as it densifies and contracts with time. Solid lines indicate bias for first year ice, dashed lines for multiyear ice assuming fixed densities of 916.7 and 882 kg m$^{-3}$ respectively. (c) Monthly thickness bias introduced by conventional and erroneous use of Eq. (6) when calculated using W99 density and depth distributions. Pixels are only displayed where sea ice type is known in all years 2010-2018. Black line indicates region where multiyear ice is present in over 50% of years. Monthly averages derived from years 2010-2018.

erroneous interpretation of Eq. (6). Were AWI to use a similar reference density to CPOM's 300 kg m$^{-3}$ with Eq. (6), their reference $\delta h$ would be $0.19Z$, contrasting starkly with CPOM's $0.25Z$.

The decision to use a fixed snow density for the wave-speed propagation correction throughout the winter introduces biases of its own with regard to the rate of thermodynamic growth; this is discussed in the next section.

## 3 Impact of Seasonal Snow Density Evolution on the Radar Wave Propagation Correction

Despite recent developments in pan-Arctic scale snow density modelling (Petty et al., 2018b), the Arctic snow density distribution remains poorly constrained in time and space. Because of this, representative values for pan-Arctic average snow density are often combined with the snow depth distributions from W99 to calculate the radar wave propagation correction (Kurtz et al., 2014; Hendricks et al., 2016; Tilling et al., 2018).

This constant value contrasts with the ubiquitous inclusion of density evolution in the adjustment to an ice floe's hydrostatic equilibrium due to the weight of overlying snow. A density evolution curve was derived from W99 by Kwok and Cunningham (2008) and implemented in sea ice thickness estimates derived from ICESat and CryoSat-2 (Kwok et al., 2009; Kwok and Cunningham, 2015). It is notable that Kwok and Cunningham (2015) include density evolution in both their calculation of the propagation correction and the adjustment to hydrostatic equilibrium.

To investigate the impact of an evolving snow density on the propagation correction, we calculated the propagation correction over Arctic sea ice by two methods: The control method used a fixed reference density in the wave speed correction (i.e. 300 kg m$^{-3}$) as done by CPOM and AWI. The other method incorporated a rate of snow densification obtained from W99 in the Central Arctic Basin.

The control method used the parameters employed by Tilling et al. (2018) producing a radar wave speed in snow of 2.4×10$^8$ m s$^{-1}$ corresponding to a reference density of 300 kg m$^{-3}$ when converted using Eq. (10). As discussed in Sect. 2, estimates of absolute sea ice thickness are sensitive to the choice of reference snow density. However, the estimated rate of thermodynamic growth (the focus of this section) is more responsive to the density's time derivative, which for a fixed value ($\rho_s = 300$ kg m$^{-3}$) is zero. As such, our results with respect to growth rate are applicable to different reference densities such as those used by AWI (350 kg m$^{-3}$) and the NASA Goddard Space Flight Center (320 kg m$^{-3}$; Kurtz et al., 2014).

For the 'evolving' method, we calculated a representative winter (Oct-Apr) densification rate using the average densification rate of snow over the Arctic Ocean given by W99. This was found to be approximately +6.50 kg m$^{-3}$ per month. The October starting density was taken as the spatial average of the W99 October density field over the same region - this choice served to minimise sea ice thickness differences at the start of the growth season and better enable comparison of growth rate. Snow density in the 'evolving' method can therefore be written as:

$$\rho_s = 6.50t + 274.51 \tag{11}$$

Where $t$ represents the number of months since October.

The W99 snow density evolution of five Arctic regions were also examined and found to be similar to the basin-wide rate, with the exception of the Laptev Sea which shows only a small (but positive) seasonal densification rate (Fig. S3). As in Sect. 2, we halved the W99 snow depths over FYI and only analysed the Central Arctic basin where W99 is considered most reliable.

When the evolving density shown in Eq. (11) was included in our calculation of the radar wave propagation correction, we found sea ice thickness to grow on average by an extra 10.1 cm between October and April over MYI. This corresponds to an extra 1.7 cm per month when compared to a fixed $f(c_s)$ of 0.25Z. Density evolution caused FYI to grow an extra 6.4 cm over the same time period, corresponding to an extra 1.1 cm per month.

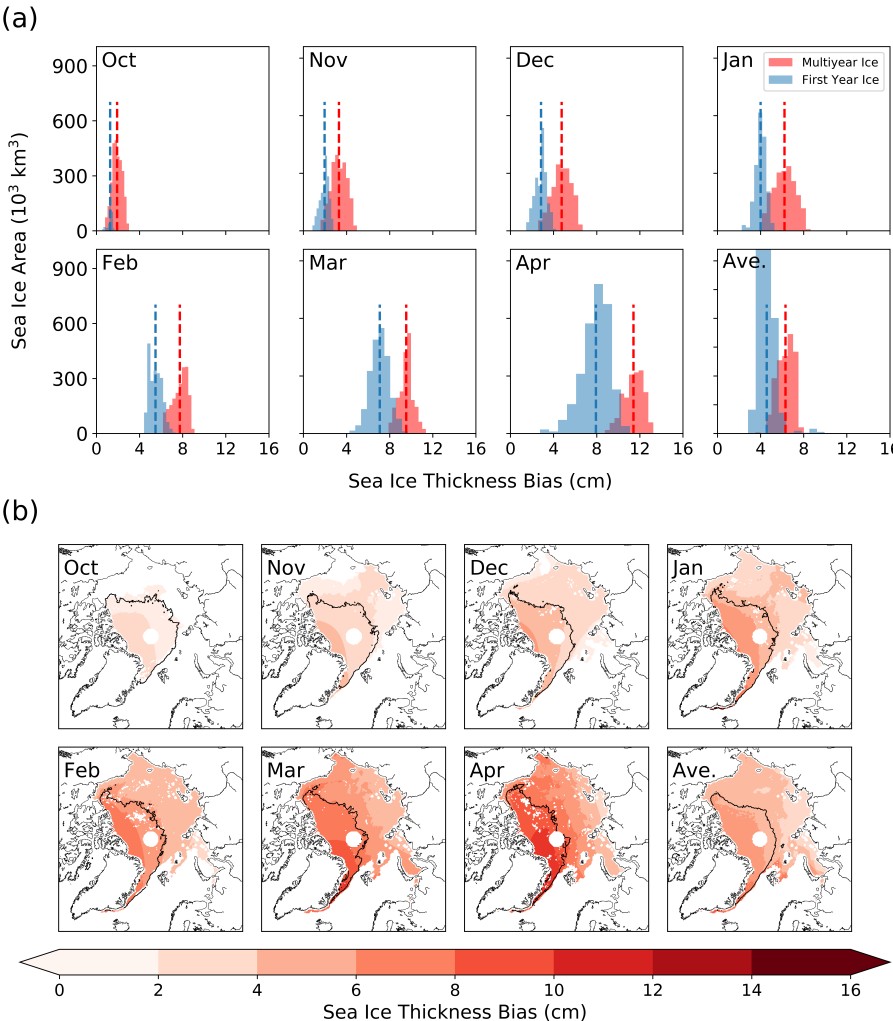

**Figure 2.** Monthly biases in sea ice thickness due to the effect of ignoring snow densification in calculating propagation correction (a) Spatially averaged histograms indicating the area of ice subjected to a given bias. Data separated into pixels that feature MYI for that month in more/less than 50% of years 2010-2018. Pixels that typically feature MYI experience greater bias in all months, largely due to halved W99 snow depths over FYI. (b) Sea ice thickness bias introduced by use of a static snow density in the calculation of the propagation correction. Pixels are only displayed where sea ice type is known in all years 2010-2018, so bias is not displayed in some areas of ambiguous ice type. Black line indicates region where MYI is present in over 50% of years.

Given the poor state of knowledge concerning the current distribution of pan-Arctic snow densities and the difficulty in collecting in-situ data, we cannot conclude whether this increased growth should correspond to higher-than-previous thicknesses at the end of winter or lower-than-previous thicknesses at the start of winter. Put another way, in this section we show a systematic bias in the thermodynamic growth rate rather than absolute ice thickness values.

Having illustrated the effect of snow densification on the radar wave propagation correction, we now justify its inclusion.

While the absolute values for regional mean densities have conceivably changed since the data was collected for W99, it remains almost certain that snow density still increases over winter for the majority of the Arctic basin as documented in W99. Furthermore, the rate of snow densification shown in W99 is likely now underestimated, with field observations indicating densification rates of >20 kg m$^{-3}$ per month on FYI (Langlois et al., 2007) and FYI now occupying significantly more of the Arctic basin than in the 1954-91 period over which W99 was compiled (Stroeve and Notz, 2018). While significant uncertainty in the true densification rate exists, effectively setting the rate to zero for the radar wave propagation correction introduces a systematic bias in sea ice thickness calculations.

Finally, commonly used products (e.g. Tilling et al., 2018; Hendricks et al., 2016) have included a seasonally evolving snow density in the 'snow loading correction' (for change in the hydrostatic equilibrium of the floe due to the weight of snow cover), which features a very similar sensitivity to uncertainty in snow density (Fig. S4).

## 4  Discussion

### 4.1  Different Fixed Densities

To further explore this issue, we calculated the expected difference between sea ice thickness estimates from CPOM and AWI introduced by their usage of $\delta h = 0.25Z$ and $\delta h = 0.22Z$ respectively. Since the difference in $\delta h$ is partially due to different choices of a representative snow density, resulting sea ice thickness differences cannot be seen as bias from a true value until Arctic snow densities are better constrained. This variation is superimposed on the bias introduced by fixed snow densities discussed above. We find that CPOM's higher value for $f(c_s)$ produces a higher mean MYI thickness of 5 cm in November, growing to 7 cm by April. 16% of MYI exhibited a difference of > 8 cm. For FYI, the mean difference is 2.8 cm in November and grows to 4.7 cm by April (Fig. S5).

### 4.2  Comparison to Radar Freeboards

To investigate these biases further, we compare them by converting pan-Arctic CryoSat-2 radar freeboard retrievals from late 2010 to early 2018 (processed under the assumption of a lognormal ice roughness distribution (Landy et al., 2019)) to estimates of sea ice freeboard using:

(a) Equation (5) versus Eq. (6) (with conventional, erroneous interpretation) using the depth and density fits from W99

(b) A monthly evolving density versus the fixed density used in Hendricks et al. (2016), both with spatially constant density across the Arctic basin

We find that the bias introduced by the erroneous interpretation of Eq. (6) remains relatively constant as a fraction of the sea ice freeboard at around 6% (despite increasing in an absolute sense) (Fig. 3a).

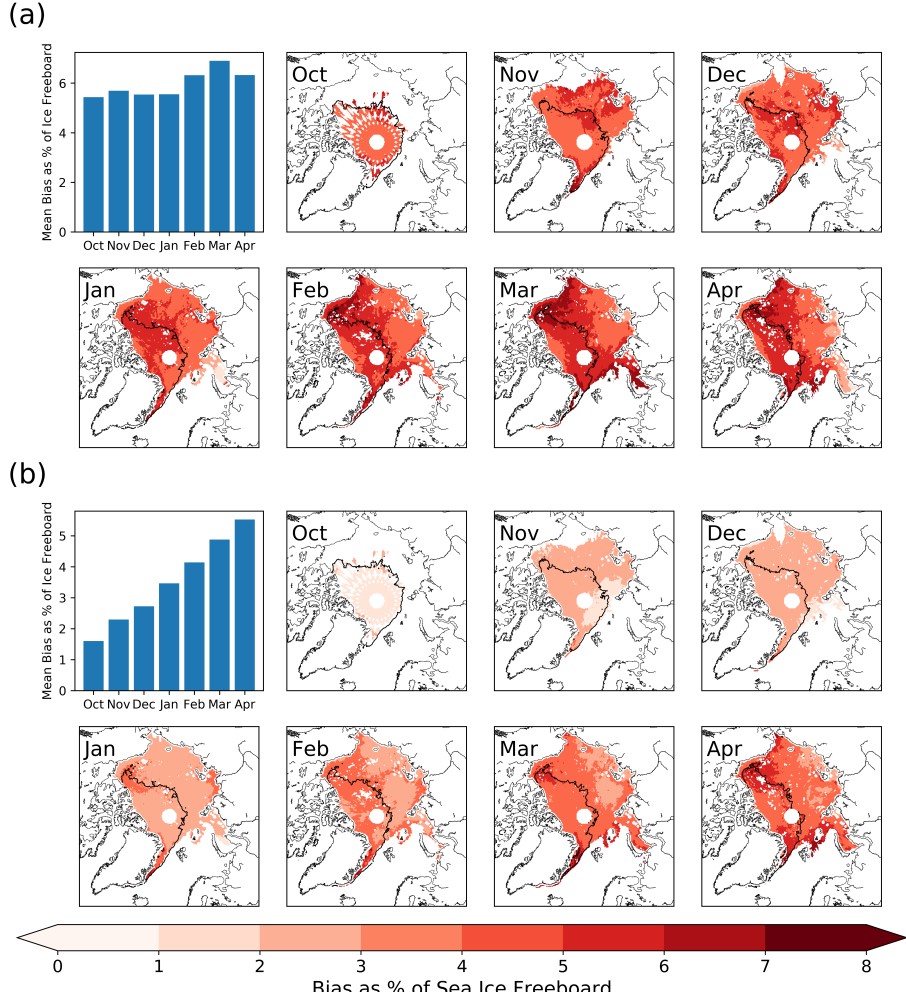

**Figure 3.** Percentage bias in sea ice freeboard. The bias induced by two effects was compared to the radar freeboards proccessed using the assumptions of Landy et al. (2019). (a) Percentage bias introduced by the use of Eq. (5) vs Eq. (6) when combined with the W99 fits for depth and density. As a fraction of the growing ice freeboard, biases remain relatively constant, indicating they grow at the same rate. (b) Percentage bias introduced by an evolving snow density derived from W99 data. This bias increases as a fraction of the ice freeboard from 2.3% to >6%, indicating that thermodynamic growth rates are underestimated.

We find that the bias indtrouced by the assumption of a non-evolving snow density (in calculation of the propagation correction) grows throughout the season relative to the sea ice freeboard and in an absolute sense. The bias grows from 2.3% to 6% of the ice freeboard (Fig. 3b), indicating that the growth rate is underestimated when a fixed density is assumed.

## 4.3 Incomplete Radar Wave Penetration of the Snowpack

The biases introduced in this analysis are derived based on the common assumption that Ku-band radar waves penetrate the entire snowpack. However, in-situ studies of Antarctic snow on sea ice indicate that snow with significant morphological features can scatter the radar above the snow-ice interface (Willatt et al., 2009). Airborne investigations during the CryoVEx and N-ICE2015 campaigns also revealed elevated dominant scattering horizons (Willatt et al., 2011; King et al., 2018). Furthermore, snow salinity has also been shown to elevate the dominant scattering horizon from the snow-ice interface. Nandan et al. (2017) found the horizon to be elevated by 7 cm based on FYI data from the Canadian Arctic.

Radar wave scattering from a horizon above the ice-snow interface introduces an overestimating bias on sea ice freeboard and thickness. The size of this bias is potentially larger than those discussed above, and may be dominant in determining the sign of the overall bias. If this is the case and sea ice thickness is overestimated overall, fixing the underestimating biases discussed in this analysis would shift estimates further away from the true value. As such, while improving the realism of the retrieval algorithm, the results may not become more accurate.

## 4.4 Snow Depth Decline Since W99 Collection

The climatology assembled by Warren et al. (1999) was collected from drifting ice stations largely over MYI in the period 1954-91. Since then the average age of MYI has declined and freeze-ups have become increasingly delayed (Stroeve and Notz, 2018). This has had the effect of decreasing snow depth over MYI (Webster et al., 2014). While W99 has been modified to better apply over FYI using comparatively recent Operation Ice Bridge data (e.g. Laxon et al., 2013; Webster et al., 2014), this has not been similarly carried out for MYI snow depths in this analysis or other publicly available products. As such, the snow depths conventionally used for thickness retrievals are likely overestimates over MYI and this introduces an overestimating bias on freeboard and sea ice thickness. This would add to the effect described in Sect. 4.3, where fixing underestimating biases may not make the overall estimate closer to the truth.

Furthermore, lower snow depths and/or incomplete radar wave penetration of the snowpack would decrease the magnitude of the biases described here (as Eq. (8) and Eq. (9) both scale linearly with snow depth).

## 4.5 Broader Implications

Sea ice thickness is closely tied to sea ice volume, a sensitive indicator of climate change but also a quantity of major interest for the modelling community. The thickness underestimates highlighted in Sect. 2 have some impact on total sea ice volume, although this is well within the currently large uncertainty bounds. Nonetheless, we argue that these uncertainty bounds have been systematically biased through conventional use of Eq. (6) in some products.

In addition, the fact that these biases grow over winter means the seasonal growth rate is also biased through conventional use of Eq. (6). While the rate of winter sea ice growth is still uncertain and interanually variable, the use of a fixed, seasonally-constant value for the snow density will bias growth rates low.

Accurate characterisation of thermodynamic growth is important to a variety of systems. A higher growth-rate will impact the surface salinity balance as more freshwater than previously estimated is locked up in sea-ice during thermodynamic growth and then ejected to the mixed-layer when ice melts in summer. The rate of sea ice growth is an important variable in the characterization of the negative conductive feedback (thin ice thickens faster: Stroeve et al., 2018; Petty et al., 2018a). Finally, end of winter sea ice thickness moderates subsequent light transmittance through the ice, impacting under-ice ecosystems and related geochemical processes (Nicolaus et al., 2012).

Sea ice thickness products featuring the misinterpretations of Eq. (6) have fed several forecast and reanalysis models (e.g. Xia and Xie, 2018). Thickness products featuring a constant-density assumption built into the propagation correction are near-ubiquitous (with the exception of Kwok and Cunningham (2015)) and have also fed forecast and reanalysis models (e.g. Yaremchuk et al., 2019; Blockley and Peterson, 2018). While these biases may be small compared to the effects of partial radar wave penetration into the snowpack, they are simply remediable. We suggest this is done before further work is undertaken to estimate the extent of and incorporate the effects of partial radar wave penetration into the snow cover.

## 4.6 Summary

We investigated two conventional methods for correcting radar altimetry derived sea ice freeboard retrievals for slower radar wave propagation in snow. We found that a commonly used treatment (conventional use of of Eq. (6)) for this correction introduces an initial and seasonally-increasing underestimating bias on sea ice thickness from October through to April. While most commonly-used products then transform this bias (where present) by choosing a fixed snow density, we find underestimation of April sea ice thickness to exceed 15 cm over some multiyear ice when this treatment is applied in conjunction with the snow density climatology from Warren et al. (1999).

We also investigated the impact of assuming a seasonally-fixed snow density on the radar wave propagation correction. While uncertainties in the absolute value of Arctic snow density preclude any conclusion about whether sea ice thickness is being under- or overestimated in this respect, this treatment is found to introduce an underestimating bias on the thermodynamic growth rate of multiyear ice of $\sim$1.7 cm per month leading to a $\sim$10.1 cm bias over the October-April period.

While these biases on sea ice thickness (Sect. 2) and growth rate (Sections 2 & 3) retrievals are small compared to the total uncertainty, they are systematic and influence the uncertainty bounds. These biases also propagate into derived products and model projections, calibrations and reanalyses.

*Code and data availability.* https://github.com/robbiemallett/snow_density_assumptions

*Author contributions.* RDCM carried out the analysis and wrote the manuscript, with continued input from all authors. In addition to manuscript input, JCS and JCL contributed data to aid analysis and MCT contributed to the processing code.

*Competing interests.* The authors declare no competing interests.

*Acknowledgements.* This work was funded primarily by the London Natural Environmental Research Council Doctoral Training Partnership grant (NE/L002485/1). JCL acknowledges support from the European Space Agency Living Planet Fellowship 'Arctic-SummIT' under grant ESA/4000125582/18/I-NS and the Natural Environmental Research Council Project 'Diatom-ARCTIC' under Grant NE/R012849/1. MT acknowledges support from the European Space Agency in part by project 'Polarice' under grant ESA/AO/1-9132/17/NL/MP and in part by the project 'CryoSat + Antarctica' under Grant ESA AO/1-9156/17/I-BG.

The authors would also like to thank Dr. Ron Kwok, Dr. Rasmus Tonboe and Dr. Rachel Tilling for their constructive comments which helped improve and clarify the manuscript.

240

245

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
