# Peer review of "Brief Communication: Conventional assumptions involving the speed of radar waves in snow introduce systematic underestimates to sea ice thickness and seasonal growth rate estimates"

_The Cryosphere, 2019_

## Referee Comment (RC1)

General Comments:

This brief communication focuses on two topics.

The first topic discusses the calculation of path length deviations ($\delta_p$), in a snow layer of thickness, $h_s$, using the following equations:

$$\delta_p = h_s(1 - \frac{c_s}{c}) \qquad (1)$$

$$\delta_p = h_s(\frac{c}{c_s} - 1) \qquad (2)$$

where $c$ and $c_s$ are the speed of light in vacuo and speed of light in snow with a given bulk density. Equation (2), in the manuscript, is the proper way to calculate the simple quantity. Equation (1) was first introduced in Section 3b of *Kwok and Cunningham* [2015] – a transcription error – and corrected in subsequently publications that utilizes path length calculations: see *Kwok and Markus* [2017] and *Kwok and Kacimi* [2018].

While it is useful (for the community) to note the impact of using Equation (1), the reviewer (and author of *Kwok and Cunningham* [2015]) feels and requests that– if this article were to be published – it should be noted that the equations are correctly written in the subsequent publications listed above. Even though the induced errors are small, this brief communication should be useful to readers who were not aware of this error.

The second topic has to do with the impact of thickness retrievals from freeboard using a fixed bulk snow density rathan than a seasonally variable snow density (or simple densification over time in this case) on retrievals. The comparisons in thickness differences (and therefore growth rates) using the variable densities are interesting and useful illustrations, especially the separate analyses of first-year and multi-year ice. The authors neglected, however, to note that *Kwok and Cunningham* [2008] first discussed seasonally varying snow density, and have used a modified seasonally varying snow density model in all of their freeboard and thickness calculations from ICESat [*Kwok et al.*, 2009] and CryoSat-2 [*Kwok and Cunningham*, 2015] data sets. It is appreciated that the authors note that varying densities, though far from perfect, have been discussed though not in the same manner as that here, and are being used in thickness calculations.

I have no problem with the publication of this brief note after the minor revisions requested.

Ron Kwok

References

Kwok, R., G. Cunningham, M. Wensnahan, I. Rigor, H. Zwally, and D. Yi (2009), Thinning and volume loss of the Arctic Ocean sea ice cover: 2003-2008, *J. Geophys. Res.*, *114*, doi:10.1029/2009JC005312.

Kwok, R., and G. F. Cunningham (2008), ICESat over Arctic sea ice: Estimation of snow depth and ice thickness, *J. Geophys. Res.*, *113*(C8), C08010, doi:10.1029/2008jc004753.

Kwok, R., and G. F. Cunningham (2015), Variability of Arctic sea ice thickness and volume from CryoSat-2, *Phil. Trans. R. Soc. A*, *373*(2045), doi:10.1098/rsta.2014.0157.

Kwok, R., and S. Kacimi (2018), Three years of sea ice freeboard, snow depth, and ice thickness of the Weddell Sea from Operation IceBridge and CryoSat-2, *Cryosphere*, *12*(8), 2789-2801, doi:10.5194/tc-12-2789-2018.

Kwok, R., and T. Markus (2017), Potential basin-scale estimates of Arctic snow depth with sea ice freeboards from CryoSat-2 and ICESat-2: An exploratory analysis, *Adv. Space Res.*, doi:10.1016/j.asr.2017.09.007.

---

## Referee Comment (RC2) · Rasmus Tonboe (Referee) · 4 Oct 2019

Review of "Brief Communication: Conventional assumptions involving the speed of radar waves in snow introduce systematic underestimates to sea ice thickness and seasonal growth rate estimates" by Mallett et al.

General comments

[Figure]

The MS is a welcome contribution to the ongoing discussion of uncertainties in sea ice thickness estimates derived from satellite radar altimeter freeboards. Several groups are processing radar altimeter data for sea ice thickness using different procedures. However, the list of systematic uncertainties which are not or only partly corrected for in the estimation of ice thickness is long because the variables that we need for correction are not well constrained, e.g. snow depth, density, surface roughness, salinity, snow grain size and ice density, roughness, the water density (Tonboe et al., 2010; Nandan et al., 2017; Alexandrov et al., 2010). The sub-footprint spatial distribution of height and backscatter on the sea-ice floe leading to preferential sampling is also important (Tonboe et al., 2010). Some of these variables such as the snow depth is affecting the radar scattering horizon and the snow-ice interface in opposite directions so that the correction for one and not the other may lead to even larger errors than doing nothing. Here I see the correction of the range for the propagation speed of microwaves in the snow to be related to the scattering horizon depth variability.

However, the magnitude of the range correction described in this MS is probably overestimated because there is evidence that the scattering horizon is not synonymous with the snow ice interface (Armitage and Ridout, 2015). The scattering horizon is more likely within the snowpack also on first-year ice because the first-year ice snow cover may be saline thus preventing penetration into the bottom snowpack (Nandan et al., 2017). I think that a short discussion of that should be included. Also today's snow depth compared to the modified Warren climatology which is used for estimating the magnitude of the range correction should be included in the discussion.

Specific comments

P1, L3: This implies that that the scattering horizon is synonymous with the snow-ice interface. However there is evidence that the scattering horizon is above the snow ice interface especially if the snow is saline. This depth (scat. horiz.) is not well known, so how to apply the correction? P1,L6: "winter ice" is sometimes synonymous with "first-year ice", move "in winter" to the end of the sentence to avoid confusion. P1, L18:
less snow gives more potential for ice growth, increasing temperatures the opposite. This sentence is contradicting. P6, L135:The NP is normally not covered by satellites and so it is not a good spot for comparison or verification. P10, L197: How do you know the depth of penetration?

Alexandrov, V., Sandven, S., Wahlin, J., and Johannessen, O. M.: The relation between sea ice thickness and freeboard in the Arctic, The Cryosphere, 4, 373–380, https://doi.org/10.5194/tc-4-373-2010, 2010. Armitage, T., A. Ridout. Arctic sea ice freeboard from AltiKa and comparison with Cryosat-2 and operation IceBridge. Geophysical Research Letters 42, 6724-6731, 2015. Nandan, V., T. Geldsetzer, J. Yackel, M. Mahmud, R. Scharien, S. Howell, J. King, R. Ricker, B. Else. Effect of Snow Salinity on CryoSat‐2 Arctic First‐Year Sea Ice Freeboard Measurements. Geophysical Research Letters https://doi.org/10.1002/2017GL074506, 2017. Tonboe, R. T., L. T. Pedersen, and C. Haas. Simulation of the Cryosat-2 satellite radar altimeter sea ice thickness retrieval uncertainty. Canadian Journal of Remote Sensing 36(1), 55-67, 2010.

---

## Referee Comment (RC3) · Rachel Tilling (Referee) · 29 Oct 2019

This study focuses on the impact of snow density on the radar propagation correction applied to sea ice freeboard, and subsequent estimates of sea ice thickness and growth rate. On the whole I thought the paper was thorough and well-written, and will be of interest to the sea ice remote sensing community. However, I have a few comments that need to be addressed before publication.

**Main comments**

1.) The authors' statements about improving the accuracy of sea ice thickness estimates are simplistic and misleading. In the abstract they state that "Correcting these biases would improve the accuracy of sea ice thickness products" and this is echoed throughout the text. This conclusion doesn't account for opposing biases that also exist. For example, *Nandan et al.* [2017] found that saline snow on first-year ice decreases the radar penetration depth and increases the main scattering horizon by ~7 cm. The impact of the salinity bias on sea ice thickness estimates is opposite to the one presented in this study, and of a greater magnitude. Therefore, to improve the accuracy of sea ice thickness estimates we require an in-depth analysis of all possible biases. The authors should include a balanced discussion of other biases (some of which will have the opposite effect of the one discussed here), and address the fact that correcting only one of these biases could actually be detrimental to the accuracy of sea ice thickness estimates. In any case, it's not possible to say that any correction definitely "would" improve sea ice thickness estimates without independent evaluation of the corrected thickness dataset.

2.) It should be clearer that the study is only concerned with the impact of evolving snow density on the radar propagation correction, and not the conversion of sea ice freeboard to thickness (for which all groups apply an evolving snow density). This is suitably explicit in the title of section 3 and a couple of places in the text, but not throughout. Please check.

**Minor comments**

Introduction: Unnecessarily dense with information. The first two paragraphs could be condensed and combined.

P1L6: Rearrange for absolute clarity that 15 cm applies to sea ice thickness, not growth rate

P2L35: Reference needed

Figure 1 (a) and (b): Larger text for numbers and y-axis labels

P8L154: "…**effectively** setting the rate to zero **for the radar range correction** introduces…"

P8L155-157: Again, make it clear that these calculations do account for seasonal variation in snow density, even though they will still be sensitive to uncertainties in the density assumptions.

**References**

Nandan, V., T. Geldsetzer, J. Yackel, M. Mahmud, R. Scharien, S. Howell, J. King, R. Ricker, and B. Else (2017), Effect of Snow Salinity on CryoSat-2 Arctic First-Year Sea Ice Freeboard Measurements, *Geophysical Research Letters*, *44*(20), 10,419-410,426, doi:10.1002/2017gl074506.

---

## Author Comment (AC1) · 20 Nov 2019

Responses to all reviewers and proposed changes are provided in the pdf supplement.

Please also note the supplement to this comment:
https://www.the-cryosphere-discuss.net/tc-2019-198/tc-2019-198-AC1-supplement.pdf

---

## Author Comment (AC2) · 20 Nov 2019

Responses to all reviewers and proposed changes are provided in the pdf supplement.

Please also note the supplement to this comment:
https://www.the-cryosphere-discuss.net/tc-2019-198/tc-2019-198-AC2-supplement.pdf

---

## Author Comment (AC3) · 20 Nov 2019

Responses to all reviewers and proposed changes are provided in the pdf supplement.

Please also note the supplement to this comment:
https://www.the-cryosphere-discuss.net/tc-2019-198/tc-2019-198-AC3-supplement.pdf

---

## Author Response (AR1)

We thank Dr. Ron Kwok, Dr. Rasmus Tonboe and Dr. Rachel Tilling for their insightful and constructive comments and we describe our changes below.

A 'track changes' manuscript is appended to this response. Line numbers used in our responses refer to this manuscript.

For each reviewer we reproduce their comments in blue and our responses in black.

**Response to Reviewer 1, Dr Ron Kwok:**

Equation (2), in the manuscript, is the proper way to calculate the simple quantity. Equation (1) was first introduced in Section 3b of Kwok and Cunningham [2015] – a transcription error – and corrected in subsequently publications that utilizes path length calculations: see Kwok and Markus [2017] and Kwok and Kacimi [2018]. While it is useful (for the community) to note the impact of using Equation (1), the reviewer (and author of Kwok and Cunningham [2015]) feels and requests that– if this article were to be published – it should be noted that the equations are correctly written in the subsequent publications listed above.

We are happy to include this acknowledgement and have included the additional citations (Kwok and Markus, 2017; Kwok and Kacimi, 2018) on line 64. To better acknowledge this transition, we have rephrased lines 64-65 from "Some authors … have used this formulation" to "Some works ..."

The authors neglected, however, to note that Kwok and Cunningham [2008] first discussed seasonally varying snow density, and have used a modified seasonally varying snow density model in all of their freeboard and thickness calculations from ICESat [Kwok et al., 2009] and CryoSat-2 [Kwok and Cunningham, 2015] data sets. It is appreciated that the authors note that varying densities, though far from perfect, have been discussed though not in the same manner as that here, and are being used in thickness calculations.

We are again happy to include this discussion with the citations suggested. This has been done in lines 124-127 and line 229.

**Response to Reviewer 2, Dr Rasmus Tonboe:**

Some of these variables such as the snow depth is affecting the radar scattering horizon and the snow-ice interface in opposite directions so that the correction for one and not the other may lead to even larger errors than doing nothing. Here I see the correction of the range for the propagation speed of microwaves in the snow to be related to the scattering horizon depth variability.

However, the magnitude of the range correction described in this MS is probably overestimated because there is evidence that the scattering horizon is not synonymous with the snow ice interface (Armitage and Ridout, 2015). The scattering horizon is more likely within the snowpack also on first-year ice because the first-year ice snow cover may be saline thus preventing penetration into the bottom snowpack (Nandan et al., 2017). I think that a short discussion of that should be included.

This is a significant point which is worthy of acknowledgement in the paper, and is similar to the first point raised by Reviewer 3 (Dr Rachel Tilling). We have now ammended the manuscript, principally adding Subsection 4.3 (lines 190 – 201). Here we state that

remedying the biases analysed may move sea ice thickness estimates further from their true (and presently unknown) values. We have also ammended line 9 (in the abstract) to remove the implication that fixing these biases will definitely improve the accuracy of estimates.

While it was stated in the paper that this work uses the assumption of full radar wave penetration, the impact of this assumption on the results was not stated and is now summarised in lines 211-212.

Also today's snow depth compared to the modified Warren climatology which is used for estimating the magnitude of the range correction should be included in the discussion.

This discussion has now been included in Subsection 4.4 (lines 202-212). Since snow depths over MYI are likely lower now than in the modified W99, sea ice thickness is likely overestimated in this regard. This introduces a similar issue to that discussed directly above.

Specific Comments

P1, L3: "This implies that that the scattering horizon is synonymous with the snow-ice interface. However there is evidence that the scattering horizon is above the snow ice interface especially if the snow is saline. This depth (scat. horiz.) is not well known, so how to apply the correction?"

We have ammended line 3 to clarify that the assumption of full snowpack penetration is invoked in publicly available sea ice thickness products. However, we now acknowledge in Subsection (4.3) that this assumption has not held in numerous investiagtions (e.g. Nandan et al., 2007; Willatt, et al., 2009; Willatt, et al., 2011; King et al., 2018).

We believe the work in this manuscript to qualitatively hold for lower snow penetration depths induced by a raised scattering horizon, although the size of the biases would be reduced. We have now pointed this out in lines 211-212. In this case, the correct equation for the propagation correction should still be selected and seasonal densification would still take place and should be accounted for. To address the reviewer's point, we now discuss the effect of an elevated scattering horizon in Section 4.3.

P1,L6: "winter ice" is sometimes synonymous with "first-year ice", move "in winter" to the end of the sentence to avoid confusion.

We accept this suggestion and have rearranged the sentence accordingly.

P1, L18: less snow gives more potential for ice growth, increasing temperatures the opposite. This sentence is contradicting.

In combination with feedback from Reviewer 3 (Dr. Rachel Tilling) that this paragraph is unnecessary long and dense with information, we have removed this sentence entirely.

P6, L135:The NP is normally not covered by satellites and so it is not a good spot for comparison or verification.

We have recalculated a representative densification rate based on a spatial average of the Arctic Basin shown in supplementary figure (S2). This newly calculated rate is slightly

higher than that calculated for the North Pole (6.50 vs 6.45 kgm$^{-3}$/year), and we have repeated our analysis and updated our figures with this slightly higher value.

P10, L197: How do you know the depth of penetration?

Currently the depth of penetration is not well known and as such is assumed to be total in publicly available sea ice thickness products. To avoid the implication that radar range estimates can be simply corrected for this issue, we have ammended this sentence to read:

"We suggest this is done before further work is undertaken *to estimate the extent of and incorporate the effects of* partial radar wave penetration into the snow cover"

**Responses to Reviewer 3, Dr Rachel Tilling:**

Major Comments

"The authors' statements about improving the accuracy of sea ice thickness estimates are simplistic and misleading. In the abstract they state that "Correcting these biases would improve the accuracy of sea ice thickness products" and this is echoed throughout the text. This conclusion doesn't account for opposing biases that also exist."

The reviewer highlights several features of SIT retrieval algorithms that introduce overestimating biases. If SIT is indeed overestimated on the whole, then correcting the underestimating biases highlighted in this manuscript will make SIT estimates larger, further removing them from the true value. We therefore agree that the statement "Correcting these biases would improve the accuracy of sea ice thickness products" (L9) is incorrect if this is the case.

We have rephrased the final sentence of the abstract (L9) from:

"Correcting these biases would improve the accuracy of sea ice thickness products, which feed a wide variety of model projections..."

To:

"Correcting these biases would impact a wide variety of model projections, calibrations, validations and reanalyses."

We have also included a new subsection (4.3) to discuss the overestimating biases introduced by incomplete radar wave penetration of the snowpack and to acknowledge that if SIT is currently overestimated then fixing these biases may not make SIT retrievals closer to the true value.

Finally, we have changed the wording in the Summary (Subsection 4.6). Rather than referring to how much sea ice thickness is underestimated, we now refer to the biases that are introduced by the treatments examined in our analysis (lines 102, 208, 238 and 244).

It should be clearer that the study is only concerned with the impact of evolving snow density on the radar propagation correction, and not the conversion of sea ice freeboard to thickness (for which all groups apply an evolving snow density). This is suitably explicit in the title of section 3 and a couple of places in the text, but not

throughout.

We have now stated this explicitly in lines 128 and 164.

Minor Comments

Introduction: Unnecessarily dense with information. The first two paragraphs could be condensed and combined.

We have removed information from the first two paragraphs and combined them. This is visible in lines 14-22 of the track changes document.

P1L6: Rearrange for absolute clarity that 15 cm applies to sea ice thickness, not growth rate.

We have rearranged and added the value for growth rate bias (to distinguish from absolute SIT bias).

P2L35: Reference needed

This was an error and the propagation correction is in fact of comparable magnitude (ratio 0.25 : 0.29 for 300kgm$^{-3}$ snow). This has been ammended and illustrated in Section 1 of the supplement.

Figure 1 (a) and (b): Larger text for numbers and y-axis labels

This figure has been updated with larger x and y axis labels and ticks.

P8L154: "…**effectively** setting the rate to zero **for the radar range correction** introduces…

This line has now been ammended as suggested.

P8L155-157: Again, make it clear that these calculations do account for seasonal variation in snow density, even though they will still be sensitive to uncertainties in the density assumptions.

Line 166 now includes clarification that the evolving snow density is on a "seasonal" scale with respect to the 'snow loading correction'.

[revised manuscript text omitted]

**1 The Comparable Impacts on SIT of Snow Loading and Slower Radar Wave Propagation in Snow**

$$\text{Ice Freeboard} = \text{Radar Freeboard} + \text{Propagation Correction} \qquad \text{(Armitage and Ridout, 2015)} \qquad \text{(S1)}$$

$$h_i = h_r + h_s(c/c_s - 1) \qquad \text{(S2)}$$

5      The conversion of a given ice freeboard can be combined with a snow depth to estimate sea ice thickness:

$$SIT = h_i \frac{\rho_w}{\rho_w - \rho_i} + h_s \frac{\rho_s}{\rho_w - \rho_i} \qquad \text{(Tilling et al. 2018)} \qquad \text{(S3)}$$

Substituting Eq. (S2) into Eq. (S3)

$$SIT = h_r \frac{\rho_w}{\rho_w - \rho_i} + h_s \frac{\rho_w}{\rho_w - \rho_i}\left[\frac{c}{c_s} - 1\right] + h_s \frac{\rho_s}{\rho_w - \rho_i} \qquad \text{(S4)}$$

10   Sea Ice Thickness = Radar Freeboard Component + Propagation Correction + Snow Loading     (S5)

     Comparing the relative impacts of the Propagation Correction term and Snow Loading term is relatively simple given they share a common factor of $h_s/(\rho_w - \rho_s)$. The ratio of the two terms is therefore $c/c_s - 1$ to $\rho_s/\rho_w$. For a typical snow density of 300kgm⁻³, this ratio is 0.25 to 0.29. (Using $\rho_w = 1023.9$ and Ulaby et al. (1986) to relate $\rho_s$ to $c_s$ )

[Figure]

**Figure S1.** Three commonly used relationships between radar wave speed and snow density Hallikainen et al. (1982); Tiuri et al. (1984); Ulaby et al. (1986).

[Figure]

**Figure S2.** The region over which snow depths published in Warren et al. (1999) are generally considered reliable (Laxon et al. (2013); Kwok and Cunningham (2015)), and over which freeboards are considered in this study.

[Figure]

**Figure S3.** Winter snow densification rates for five regions and the basin-wide average. We defined the 'basin-wide area' as the shaded area in Fig. (S2). We found the basin-wide denisification rate to be roughly representative of its constituent regions apart from the Laptev, which exhibited a small but positive densification.

[Figure]

**Figure S4.** While the functional form and magnitude of expressions for the effect of snow weight and slower radar propagation are different, they have a similar error dependence on snow depth. That is to say, the percentage error introduced to the "weight correction" by snow density uncertainty is the same as that for the "propagation correction".

[Figure]

**Figure S5.** Monthly differences in sea ice thickness from the use of $\delta h = 0.22Z$ and $\delta h = 0.25Z$ for the propagation by AWI and CPOM respectively.